# Childhood appendectomy and subsequent psychiatric illness

Cecilia Arana Håkanson[1]*, Fredrik Stiger[2], Nele Brusselaers[3,4], Helene Engstrand Lilja[1]

1 Department of Women's and Children's Health, Karolinska Institute, Stockholm, Sweden, 2 Department of Surgical Sciences, Anaesthesiology and Intensive Care, Uppsala University, Uppsala, Sweden, 3 Centre for Translational Microbiome Research, Karolinska Institute, Stockholm, Sweden, 4 Global Health Institute, Antwerp University, Antwerp, Belgium

☺ These authors contributed equally to this work.
* cecilia.arana.hakanson@ki.se

## Abstract

### Aim

The appendix is considered a reservoir for the gut microbiome to restore the gut microbiota after disruptions. Therefore, removal of the appendix because of appendicitis, might result in long-term disruption of the microbiome with related health consequences. The aims of the study were to explore if there is an association between childhood appendectomy, the risk of psychiatric illness and healthcare consumption later in life.

### Methods

Individuals (N = 752) who underwent childhood appendectomy in a tertiary paediatric surgical department in Sweden were included, individually-matched to 5 non-exposed individuals (N = 3760). Data on psychiatric diagnoses, healthcare visits, and use of psychiatric drugs were collected from population-based registers. Stratified Cox and Poisson regression analyses were used, adjusted for antibiotics, proton pump inhibitors and non-steroidal anti-inflammatory drugs.

### Results

Median age at appendectomy was 11 (2–16) years and median follow-up time 15.5 (6–21) years. The risk of psychiatric illness in general (HR 1.19; 95% CI 1.04–1.37; p-value 0.013) and affective disorders (HR 1.20, 95% CI 1.01–1.42; p-value 0.038) was higher among those with appendectomy. Out- (IRR 1.20; 95% CI 1.18–1.23; p-value <0.001) and in-patient visits (IRR 1.19, 95% CI 1.10–1.28; p-value <0.001) were more common among those with appendectomy.

### Conclusion

Childhood appendectomy was associated with an increased risk of psychiatric illness and healthcare consumption.

**Data Availability Statement:** The datasets generated and analysed during the current study are not publicly available due to confidentiality. The data are available from the Swedish National Board of Health and Welfare and Swedish Ethical Review

Authority for researchers who meet the criteria for access to confidential data. Data are however available from the authors upon reasonable request and with permission of the Swedish National Board of Health and Welfare and Swedish Ethical Review Authority. Contact information: The Swedish National Board of Health and Welfare: email: registerservice@socialstyrelsen.se phone: +46-752473000 The Swedish Ethical Review Authority: email: registrator@etikprovning.se phone: +46-104750800.

**Funding:** This work was supported by H.K.H. Kronprinsessan Lovisas Förening för barnasjukvård, grant ID: 2021-00616; 2022-00686; 2023-00762 (authors: CAH, HEL) and the Gillbergska Foundation (author: CAH). The funders had no role in the conceptualization, design, data collection, analysis, decision to publish or preparation of the manuscript.

## Introduction

Appendicitis is the most common surgical emergency in the population with a lifetime risk of approximately 8% [1] and 2.5% during childhood [2]. The gold standard treatment for appendicitis in children is appendectomy. Ongoing clinical trials are evaluating whether antibiotic treatment is a safe alternative to appendectomy in non-complicated appendicitis in children [3].

The appendix was up until a few decades ago considered a structure without a recognised function and possible consequences of appendectomy were overlooked. The appendix is nowadays considered to have two main roles. First, it is a lymphoid tissue and it is the main site of IgA production in the body [4]. Second, it works as a reservoir for gut microbiota [4]. The gut microbiota is a population of microorganisms, mainly bacteria but also viruses, protozoa, fungi and archaea [5]. It can be considered a living organ in a delicate relationship with its human host. The collective genome of the gut microbiome exceeds the human over 100 fold [5]. Neurotransmitters and neurohormones such as dopamine, serotonin, norepinephrine, gamma-aminobutyric acid (GABA) and melatonin are both produced and modulated by the gut bacteria [6, 7]. The unique shape and anatomical location in the caecum, positioned away from the faecal stream, enables a protection of the microbiota within. After a disturbance of the microbiota of the gut, the microbes are given the opportunity to be re-transplanted from the appendix to the gut [4]. Removal of the appendix might be negative in terms of restoring the normal microbiota of the gut.

Appendectomy has previously been associated with a number of somatic diseases such as inflammatory bowel disease, colorectal cancer, cardiovascular disease and Parkinson's disease [8]. It was when conducting another study [9] that we, en-passant, noticed a surprisingly high proportion of psychiatric illness in the medical charts in patients that had undergone appendectomy during childhood.

Recently an association with appendectomy and psychiatric disease has been observed in two Swedish register studies [10, 11]. The first study in 2019 described an association between appendectomy and tonsillectomy and the risk of psychiatric disorders and suicidal behaviour [10]. The second study including all Swedish individuals born between 1972–1992, described an increased risk for depression, bipolar- and anxiety disorders after appendectomy in children under the age of 14. Interestingly there was a subgroup of individuals with appendicitis but without appendectomy and within this group, no increased risk was found [11].

Studies exploring the gut microbiome in patients with psychiatric illness suggest that the microbiome might be involved in the development of psychiatric disorders [12–15]. An altered faecal microbiome, has been observed in individuals with depression [12, 13], first episode psychosis [14] and bipolar disease [15].

The aim of this study was to assess if there was an association between childhood appendectomy and the risk of psychiatric illness later in early adulthood in our Swedish surgical cohort, and if there was an association in health seeking behaviour.

## Methods

### Cohort

This Swedish surgical cohort includes 752 patients, 1:5 matched to 3760 controls, designed to address the hypothesis that childhood appendectomy affects the risk of future psychiatric illness. The study cohort, the exposed group, underwent appendectomy before the age of 15 during the years 2000–2014 at the University Children's Hospital in Uppsala, Sweden. The unexposed group were randomly selected from the Swedish Drug Register (SDR) and

individually-matched for sex, year of birth and county. The unexposed group were not exposed to appendectomy from birth to the last date of follow up. The term index was used for the date of appendectomy for the exposed individuals. For the matched non-exposed individuals, index was the date of appendectomy for their matched exposed individual. Follow up time was calculated as years from index to last date of follow up, December 31st, 2020.

## Data collection

Data was collected from the Swedish national population-based health registries and received on the 14th of September 2021: Through linkage with the Swedish National Patient Registry (NPR) and the Swedish Prescribed Drug Registry (SDR), the cohort was screened for prevalence of prescribed drugs related to psychiatric illness and drugs known to affect the composition of the intestinal microbiota. The SDR contains data on all prescribed drugs dispensed at pharmacies in Sweden. The register started 1 July 2005 and data in the study was collected until December 31st 2020, hence data on drug use is limited to cases with index after this date. The Anatomical Therapeutic Chemical classification system (ATC) was used to identify drugs of interest. This included ATC codes for drugs affecting the nervous system (ATC: N05A-C; N06A-C; N07B), as well as three types of drugs known to affect the microbiota, proton pump inhibitors (PPI) [16] (ATC: A02BC), anti-infective drugs [17] (ATC: J01-J05) and non-steroidal anti-inflammatory drugs (NSAID) [18], (ATC: M01). The drugs affecting the nervous system was divided into subgroups representing different groups of psychiatric illnesses according to International Classification of Disease, tenth revision, (ICD-10), (S1 Table).

The NPR started in 1964, the register contains information about diagnoses according to the ICD and data on hospital-, and healthcare visits within the specialist care. Data from NPR was gathered from date of birth until 31st December, 2020.

## Outcome

Psychiatric illness was defined in two ways. Either through ATC codes of drugs affecting the nervous system in the SDR or the NPR for ICD codes for psychiatric disease, F00-F99. For the ATC codes subjects with at least two dispensations of drugs affecting the nervous system were included. If a subject was diagnosed in both ways, the first date was used as the date of diagnosis. Individuals with a psychiatric diagnosis or at least two dispensations of drugs affecting the nervous system before index were excluded from further analysis. An accumulated number of contacts with the specialist healthcare, including both hospitalisation and outpatient specialist visits, was gathered from the NPR. As dispensations of drugs, diagnoses and healthcare visits were gathered from records the registry dataset contains no missing data.

From a previous study of the exposed in the cohort [9] information on: date of surgery and estimation of the severity of inflammation of the appendix or perforation was known and linked to the data from the registers. The degree of inflammation was determined either by a histopathological analysis or, if missing, by an estimation from the surgeon as described in the patient's operation chart. The degree of inflammation was grouped into uncomplicated and complicated (gangrenous and perforated) appendicitis. This data was added to the register data by the National Board of Health and Welfare anonymously.

A confidentiality review was done by the registry holder Swedish National Board of Health and Welfare to confirm that the data could be disclosed without harm to the persons who has provided the data in a truly non-identifiable way.

## Statistical analyses

Categorical data were presented as frequencies or proportions. Continuous data as a median with interquartile range (IQR). Comparison of in- and outpatient visits were analysed with mixed Poisson regression presented as incidence rate ratios (IRR) and 95% confidence intervals (CI). The risk of psychiatric illness was assessed with stratified Cox proportional Hazard models, expressed as hazard ratios (HR) and 95% CI, assessing subgroups of psychiatric disorders; and taking into account the degree of inflammation. Stratified Cox regression for psychiatric illness was adjusted for use of PPI, NSAID and antibiotics within two years after appendectomy. The proportional hazards assumption was checked by using statistical tests and graphical diagnostics based on the scaled Schoenfeld residuals. All analyses were performed using R version 4.1.0.

## Ethics

The study was approved by the Regional Ethical Review Board in Uppsala, Sweden (DNR: 2020–01452). Consent was not considered necessary to obtain by the review board as the data from the registers were attained and analysed anonymously.

## Results

### Descriptive data

There were 752 exposed and 3760 non-exposed included with a median follow-up of 15.5 years (6–21). Median age at index was 11 years (2–16) and 60.6% were men (Table 1). Regarding drugs affecting the gut microbiome, 236 (63.1%) of the exposed and 458 (24.5%) of the non-exposed had received at least one prescription of anti-infective drugs within 2 years after index. PPI was prescribed to 9 (2.4%) exposed and 25 (1.3%) non-exposed, NSAID was prescribed to 17 (4.5%) exposed and 53 (2.8%) non-exposed (Table 1).

### Psychiatric illness

Psychiatric illness was identified in 34 (4.5%) exposed and 156 (4.1%) non-exposed before index. Psychiatric illness within two years after index was found in 27 (3.6%) of the exposed and in 97 (2.6%) of the non-exposed. A first occasion of psychiatric illness at least 2 years after the index was found in 239 (31.8%) exposed and 1086 (28.9%) non-exposed. The source for identifying psychiatric illness were PDR in 66 (8.8%) exposed and 347 (9.2%) non-exposed, NPR in 68 (9.0%) exposed and 280 (7.4%) non-exposed and a combination of both in 166 (22.1%) exposed and 712 (18.9%) non-exposed (Table 1).

Stratified Cox Regression gave an increased HR of 1.19 (95%CI 1.04–1.37) for psychiatric illness in exposed compared to the non-exposed. When controlling for drugs affecting the microbiome within two years after index, HR was similar 1.15 (95% CI 0.1.00–1.33) (Table 2). When looking into the different subgroups of psychiatric illness a 20% increased risk (HR 1.20 95% CI 1.01–1.42) was observed for affective disorders in exposed compared to non-exposed (Table 2).

The severity of inflammation of the appendix, complicated disease (gangrenous or perforated), was not associated with an increased risk of psychiatric illness HR 0.86 (95% CI 0.67–1.10; p-value 0.222).

### Healthcare visits

Mixed Poisson regression for outpatient visits after index showed a 20% higher rate of visits for exposed, IRR 1.20 (95% CI 1.18–1.23). For inpatient visits a 19% higher rate of visits for exposed were noted, IRR 1.19 (95% CI 1.10–1.28) (Table 3).

**Table 1. Demographic variables, descriptive statistics for psychiatric illness and use of drugs affecting the microbiome.**

| Variable | Non-exposed (N = 3760) | Exposed (N = 752) | P-value |
|---|---|---|---|
| Follow-up time in years, (median, IQR) | 15.5 (11.9–18.9) | 15.5 (11.9–18.9) | 1.000 |
| Age at time of index in years, (median, IQR) | 11 (9–13) | 11 (9–13) | 0.9682 |
| Age groups at time of index | | | |
| Preschool (1–5 years) | 220 (5.9) | 39 (5.2) | 0.6963 |
| Elementary school (6–9 years) | 962 (25.6) | 193 (25.7) | |
| Middle school (10–12 years) | 1444 (38.4) | 286 (38.0) | |
| High school (13–15 years) | 1134 (30.2) | 234 (31.1) | |
| Sex | | | |
| Male | 2280 (60.6) | 456 (60.6) | 1.0000 |
| **Psychiatric illness** | | | |
| No diagnosis | 2421 (64.4) | 452 (60.1) | 0.1053 |
| Diagnosis before index* | 156 (4.1) | 34 (4.5) | |
| Diagnosis within two years after index | 97 (2.6) | 27 (3.6) | |
| Diagnosis two years after index | 1086 (28.9) | 239 (31.8) | |
| **Diagnosis identified from** | | | |
| No diagnosis | 2421 (64.4) | 452 (60.1) | 0.0662 |
| From NPR | 347 (9.2) | 66 (8.8) | |
| From PDR | 280 (7.4) | 68 (9.0) | |
| From NPR and PDR combined | 712 (18.9) | 166 (22.1) | |
| **Prescribed drugs ** | **Non-exposed (N = 1870)** | **Exposed (N = 374)** | |
| Proton pump inhibitors | 25 (1.3) | 9 (2.4) | 0.1889 |
| Non steroidal anti-inflammatory drugs | 53 (2.8) | 17 (4.5) | 0.1153 |
| Antibiotics | 458 (24.5) | 236 (63.1) | <0.0001 |

Values in parentheses are percentages unless otherwise indicated.

P-values for categorical variables are from chi-square test. P-values for numerical variables (follow-up time, age at time of index in years) are from Mann-Whitney U-test.

* Individuals with a psychiatric diagnosis registered in NPR, or at least two dispensations of drugs affecting the nervous system in SDR before index.

**Data on prescribed drugs from 1 July 2005 and onwards, from appendectomy to two years postoperatively.

## Discussion

Clinical studies of childhood appendectomy often focus on early complications and advancements in surgical techniques. For the child, consequences beyond the postoperative period are of great importance as the child has to live and develop with possible adverse effects. The

**Table 2. Stratified Cox regression for psychiatric illness after index.**

| Psychiatric illness | N-exposed/N-non-exposed | HR | 95%-CI | P-value |
|---|---|---|---|---|
| **General** | 266 (37.0%) /1183 (32.8%) | 1.19 | (1.04–1.37) | 0.013 |
| Adjusted for Proton Pump Inhibitors, Nonsteroidal anti-inflammatory drugs and Antibiotics* | 266 (37.0%) /1183 (32.8%) | 1.15 | (1.00–1.33) | 0.046 |
| Mood disorders | 170 (22.8%) /751 (20.0%) | 1.20 | (1.01–1.42) | 0.038 |
| Anxiety disorders | 166 (22.2%) /743 (20.0%) | 1.14 | (0.96–1.35) | 0.130 |
| Psychotic disorders | 29 (3.9%) /127 (3.4%) | 1.17 | (0.78–1.76) | 0.443 |
| Neuropsychiatric | 71 (9.8%) /323 (8.8%) | 1.09 | (0.84–1.41) | 0.534 |
| Eating disorders | 21 (2.8%) /77 (2.0%) | 1.38 | (0.85–2.25) | 0.190 |

*Covariates from 1 July 2005 and onwards

**Table 3. Mixed Poisson regression for out- and inpatient visits after index.**

| Healthcare consumption | IRR | 95%-CI | P-value |
|---|---|---|---|
| **In-patient visits** | | | |
| Overall | 1.19 | (1.10–1.28) | <0.001 |
| **Out-patient visits** | | | |
| Overall | 1.20 | (1.18–1.23) | <0.001 |

appendix was up until recently considered a structure without a recognized function and the possibility of long-term consequences beyond the surgical intervention has so far been over-looked. In this study with detailed clinical records of the appendectomy we found an associa-tion between childhood appendectomy and later psychiatric illness in general and for affective illness in particular and also an increased overall healthcare consumption.

Our findings are supported by a Swedish registry study who found and increased risk of anxiety, depression and bipolar disorders after childhood appendectomy [11]. A limitation of that study was that only the NPR was used for identification of psychiatric illness. This leads to a risk of selection bias as psychiatric illness in children is often treated within the primary care and the NPR only reports diagnoses documented within the specialist care. By adding the PDR to our data, we were able to find individuals treated with drugs prescribed by the primary care. By doing this another 9.0% of exposed and 7.4% of non-exposed with psychiatric illness were identified, still the results of the two studies remained in concordance.

A possible factor confounding our result is the association between general anaesthesia and cognitive, behavioural and developmental disorders [19–21] although results from other stud-ies are conflicting [22, 23]. A Swedish registry study evaluating the risk of depressive and anxi-ety disorders after childhood appendectomy and childhood hernia surgery found an association after appendectomy but also a small but significant association after hernia surgery. However, when adjusting for mental disorders in unexposed siblings the association disap-peared after hernia surgery while persisted after appendectomy [11].

The findings in our study indicate that appendectomy was associated with an increased risk of psychiatric illness. However, the treatment of appendicitis with appendectomy also includes treatment with antibiotics per-operatively and at times pre- and/or postoperatively. Recent studies have described an association between the use of antibiotics and an increased risk of psychiatric illness [24, 25]. Interestingly a recent Danish register study observed that children who have received antibiotics were at an increased and dose-dependent risk of appendicitis [26]. In our study antibiotics were prescribed in 63.1% among exposed after index, more than double than among non-exposed (24.5%). A relatively high proportion of these prescriptions were explained by complicated appendicitis treated with broad-spectrum antibiotics after the appendectomy. Yet, in our study we did not find an increased risk of psychiatric illness after complicated appendicitis compared to uncomplicated appendicitis. Due to limited power and as the drugs were not the main exposure, we did not explore dose-response effects, yet it appears important to assess this potential association in more depth in further research, particu-larly for antibiotics. Neither did we investigate if complicated appendicitis affected specific subgroups of psychiatric illness because of the limited power when subdividing into sub-groups. Further, when attempting to control for use of antibiotics and psychiatric illness, the use of antibiotics within two years after index gave an increased HR for behavioural, emotional and affective disorders regardless of being exposed or non-exposed. For the other drugs affect-ing the microbial composition, PPI and NSAID, we did not see this effect. Whether the increased risk of psychiatric illness was associated with the removal of the appendix, the

exposure to antibiotics, the targeted infection or a combination of these remains unknown. However, in a Swedish registry study there was a group with appendicitis but without appendectomy and in that group no increased risk of psychiatric illness was observed [11] indicating an important role of the appendix itself.

Previous studies have described an association of appendectomy and conditions primarily affecting individuals beyond childhood and adolescence such as colorectal cancer, cardiovascular disease, Parkinson´s disease but also in IBD [8]. The finding in our study indicate that the general health might be affected even earlier in life as the rate of both in- and outpatient healthcare visits increased among the exposed compared to non-exposed during follow up. This increase is unlikely related to gastrointestinal problems related to the original appendectomy. One part of the increase is explained by an increase of visits within the psychiatric care. Among the exposed 246 (32.7%) had at least one visit within the psychiatric care, compared to 1044 (27.4%) of the non-exposed. Further, somatization, the expression of psychological or emotional factors as physical symptoms could also be taken into consideration as this is common in anxiety and depression disorders [27] and might contribute to an overall increase in healthcare visits. Another explanation could be that the general health is impaired after appendectomy as a number of non-gastrointestinal and gastrointestinal diseases has been associated with both appendectomy [8] and the gut microbiome [28, 29]. The effect on healthcare visits could also be confounded by a health seeking behaviour among cases. Nevertheless, in our study the number of healthcare visits before index indicate that both exposed and non-exposed were at equal general health before the appendectomy.

One of the major strengths of the study are that the medical records were individually reviewed for the cases with detailed information confirming appendicitis and the subsequent treatment and data not only data based on registers. By adding the SDR to PDR, the study also includes individuals treated only within the primary care. With the definition of psychiatric illness as at least two prescriptions of drugs to classify as a positive result we aimed to increase the accuracy of drugs being a proxy for a positive case of psychiatric illness. Also the matching of individuals by county reduced the risk of systemic variation in clinical practice and healthcare access.

One of the major limitations is that we do not have data on familiar history of psychiatric illness or other socioeconomic factors that could affect the outcome of both psychiatric illness and health seeking behavior. Further, data from PDR is limited to the start of the register in 2005 which leaves five years of this study without this extra data. Also, our definition of psychiatric illness will exclude individuals treated by talk therapy alone in the primary care.

According to data from the World Health Organization around 20% of the world´s children and adolescents have a mental health condition, depression and anxiety being two of the most common, and the second leading cause of death among 15–29-year-olds. Psychiatric illness can affect many areas in life with a loss of function impairing the ability to participate and contribute in the community [30]. As appendicitis is the most common surgical emergency during childhood, further investigation on the effect of childhood appendectomy on mental health is warranted. The findings are in favour of the conservative treatment of appendicitis whenever possible. As appendicitis is the most common surgical emergency during childhood this association could have a substantial impact on global mental health. The findings from this and previous studies on the topic should be taken into consideration when making new guidelines on how to treat uncomplicated appendicitis in children. Information about this possible consequence after appendectomy is important for the operating surgeon and the caregivers of the child to enhance the prevention, early detection and treatment of psychiatric illness.

## Conclusion

Childhood appendectomy was associated with an increased risk of psychiatric illness and healthcare consumption later in life. The association of appendectomy and psychiatric illness motivates future investigations of the microbiome being a causal factor in this spectrum of diseases.

## Supporting information

**S1 Table. ATC codes for drugs affecting the nervous system with corresponding ICD-10 codes.**
(DOCX)

## Acknowledgments

We would like to express our gratitude to Fabian Söderdahl, Statisticon AB, for his assistance with the statistical analyses.

## Author Contributions

**Conceptualization:** Cecilia Arana Håkanson, Fredrik Stiger, Nele Brusselaers, Helene Engstrand Lilja.

**Data curation:** Cecilia Arana Håkanson.

**Funding acquisition:** Cecilia Arana Håkanson, Helene Engstrand Lilja.

**Investigation:** Cecilia Arana Håkanson, Fredrik Stiger, Nele Brusselaers, Helene Engstrand Lilja.

**Methodology:** Cecilia Arana Håkanson, Fredrik Stiger, Nele Brusselaers, Helene Engstrand Lilja.

**Project administration:** Cecilia Arana Håkanson.

**Supervision:** Fredrik Stiger, Nele Brusselaers, Helene Engstrand Lilja.

**Validation:** Fredrik Stiger, Nele Brusselaers, Helene Engstrand Lilja.

**Writing – original draft:** Cecilia Arana Håkanson.

**Writing – review & editing:** Cecilia Arana Håkanson, Fredrik Stiger, Nele Brusselaers, Helene Engstrand Lilja.

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
