## [Decision Letter · Decision Letter 0]

21 Aug 2024

PMEN-D-24-00228

Childhood appendectomy and subsequent psychiatric illness

PLOS Mental Health

Dear Dr. Arana Håkanson,

Thank you for submitting your manuscript to PLOS Mental Health. After careful consideration, we feel that it has merit but does not fully meet PLOS Mental Health’s publication criteria as it currently stands. Therefore, we invite you to submit a revised version of the manuscript that addresses the points raised during the review process.

Please note that we have only been able to secure a single reviewer to assess your manuscript. We are issuing a decision on your manuscript at this point to prevent further delays in the evaluation of your manuscript. Please be aware that the editor who handles your revised manuscript might find it necessary to invite additional reviewers to assess this work once the revised manuscript is submitted. However, we will aim to proceed on the basis of this single review if possible. 

We look forward to receiving your revised manuscript.

Kind regards,

Avanti Dey, PhD

Staff Editor

PLOS Mental Health

Journal Requirements:

Additional Editor Comments (if provided):

Reviewers' comments:

Reviewer's Responses to Questions

**Comments to the Author**

1. Does this manuscript meet PLOS Mental Health’s publication criteria? Is the manuscript technically sound, and do the data support the conclusions? The manuscript must describe methodologically and ethically rigorous research with conclusions that are appropriately drawn based on the data presented.

Reviewer #1: Yes

2. Has the statistical analysis been performed appropriately and rigorously?

Reviewer #1: Yes

3. Have the authors made all data underlying the findings in their manuscript fully available (please refer to the Data Availability Statement at the start of the manuscript PDF file)?

Reviewer #1: Yes

4. Is the manuscript presented in an intelligible fashion and written in standard English?

Reviewer #1: Yes

5. Review Comments to the Author

Reviewer #1: The authors tries to find the association between childhood appendectomy and thereafter psychiatric illnesses. In terms of microbial disruption theory, the analysis target (i.e., appendectomy) is considerably reasonable. Their statistical methodology is also reasonable-stratified Cox regression etc. And the consideration for confounding factors-NSAID, PPI, Antibiotics- is quite justifiable. Also, using the national registries to select the psychiatric diagnoses is valid. In general, it seems to be qualified to be open to public. One minor critic-why aren't any considerations for the medication dosages (i.e., NSAID, PPI, Antibiotics dosages in association with thereafter psychiatric illnesses) in assessing the risks of psychiatric illnesses? Would you add the dosage-dependent relationships analysis for the manuscript?

6. PLOS authors have the option to publish the peer review history of their article (what does this mean?). If published, this will include your full peer review and any attached files.

**Do you want your identity to be public for this peer review?** For information about this choice, including consent withdrawal, please see our Privacy Policy.

Reviewer #1: No

---

## [Decision Letter · Decision Letter 1]

22 Oct 2024

PMEN-D-24-00228R1

Childhood appendectomy and subsequent psychiatric illness

PLOS Mental Health

Dear Dr. Arana,

Thank you for submitting your manuscript to PLOS Mental Health. After careful consideration, we feel that it has merit but does not fully meet PLOS Mental Health’s publication criteria as it currently stands. Therefore, we invite you to submit a revised version of the manuscript that addresses the points raised during the review process.

Please submit your revised manuscript by in **two weeks from now.** If you will need more time than this to complete your revisions, please reply to this message or contact the journal office at mentalhealth@plos.org. Please include the following items when submitting your revised manuscript:

We look forward to receiving your revised manuscript.

Kind regards,

Kizito Omona, PhD

Academic Editor

PLOS Mental Health

Journal Requirements:

Additional Editor Comments (if provided):

Your work is coming out very well. However, there are still a few areas of concern that need to be addressed before your work can be accepted. Critically look at the comment raised by one of the reviewers and use it to improve the manuscript.

Also;

1) Abstract: In the result section of abstract use the word "Results" and not Principal findings.

2) Limitations: Add a section after conclusion, that talks about all the limitations of your study.

Reviewers' comments:

Reviewer's Responses to Questions

**Comments to the Author**

1. If the authors have adequately addressed your comments raised in a previous round of review and you feel that this manuscript is now acceptable for publication, you may indicate that here to bypass the “Comments to the Author” section, enter your conflict of interest statement in the “Confidential to Editor” section, and submit your "Accept" recommendation.

Reviewer #1: All comments have been addressed

Reviewer #2: (No Response)

Reviewer #3: All comments have been addressed

2. Does this manuscript meet PLOS Mental Health’s publication criteria? Is the manuscript technically sound, and do the data support the conclusions? The manuscript must describe methodologically and ethically rigorous research with conclusions that are appropriately drawn based on the data presented.

Reviewer #1: Yes

Reviewer #2: No

Reviewer #3: Yes

3. Has the statistical analysis been performed appropriately and rigorously?

Reviewer #1: Yes

Reviewer #2: No

Reviewer #3: Yes

4. Have the authors made all data underlying the findings in their manuscript fully available (please refer to the Data Availability Statement at the start of the manuscript PDF file)?

Reviewer #1: Yes

Reviewer #2: No

Reviewer #3: Yes

5. Is the manuscript presented in an intelligible fashion and written in standard English?

Reviewer #1: Yes

Reviewer #2: Yes

Reviewer #3: Yes

6. Review Comments to the Author

Reviewer #1: No further comments.

Reviewer #2: I read with interest the manuscript entitled "Childhood appendectomy and subsequent psychiatric illness"

The introduction is comprehensive with a clear statement of the aim of the study at the end.

For appendectomy patients, it would be important to divide them into uncomplicated and complicated forms of appendicitis because there is a possibility that the course of treatment of complicated, especially perforated forms of appendicitis, can be a significant confounder.

Can you guarantee that all patients who underwent appendectomy up to the age of 15 did not have other associated diseases during the specified period?

Please add to the statistical analysis section the way you tested the normality of the data distribution.

It is unclear, in relation to table 1, whether the control group was exposed to appendectomy!?

In Table 1, p-values should be listed and next to them, indicate with a superscript which statistical test was used.

You stated the following "Individuals with a psychiatric diagnosis or at least two dispensations of drugs affecting the nervous system before the date of appendectomy were excluded from further analysis." In relation to the table, the statement is incorrect.

The discussion largely recognizes the many limitations of the study, which significantly reduce its significance of the study, and the conclusion is written rather boldly considering the numerous limitations.

In conclusion, considering the methodology and results, it is unclear who is the control group and who is the target group? Who is appendectomized and who is not? It's completely unclear.

Reviewer #3: (No Response)

7. PLOS authors have the option to publish the peer review history of their article (what does this mean?). If published, this will include your full peer review and any attached files.

**Do you want your identity to be public for this peer review?** For information about this choice, including consent withdrawal, please see our Privacy Policy.

Reviewer #1: No

Reviewer #2: No

Reviewer #3: No

---

## [Editor Report · Decision Letter 2]

12 Dec 2024

Childhood appendectomy and subsequent psychiatric illness

PMEN-D-24-00228R2

Dear Dr. Arana,

We are pleased to inform you that your manuscript 'Childhood appendectomy and subsequent psychiatric illness' has been provisionally accepted for publication in PLOS Mental Health.

Best regards,

Kizito Omona, PhD

Academic Editor

PLOS Mental Health